# Acute psychological stress does not influence joint position reproduction performance in the elbow joint

**Adam Koncz**[1,2]*, **Ferenc Köteles**[2,3], **Blanka Aranyossy**[3], **Áron Horváth**[2,3]

1 Institute of Health Promotion and Sport Sciences, ELTE Eötvös Loránd University, Budapest, Hungary,
2 Ádám György Psychophysiology Research Group, Budapest, Hungary, 3 Institute of Psychology, Károli Gáspár University of the Reformed Church in Hungary, Budapest, Hungary

* koncz.adam@ppk.elte.hu

## Abstract

### Background

Proprioceptive accuracy is an important aspect of motor functioning thus understanding how the stress response affects it can broaden our knowledge about the effects of stress on motor performance. There has been published only one quasi-experimental study on this topic to date, reporting a negative association between stress and proprioceptive accuracy. The aim of the present study was to explore whether the stress response influences proprioceptive accuracy in a randomized and controlled experimental setting.

### Method

Participants ($M_{age}$ = 20.4 yrs, $SD_{age}$ = 1.91 yrs) were randomly assigned to a stress (n = 29) and a control (n = 28) group. Psychological stress was induced via an online quiz involving time pressure and instant feedback on performance. Participants' perceived (state anxiety) and physiological (heart rate, heart rate variability, skin conductance level) stress response and proprioceptive accuracy (the active and passive version of the Joint Position Reproduction test for the elbow joint) were measured before and after the experimental manipulation.

### Results

The quiz substantially increased only participants' perceived stress however, proprioceptive accuracy was not impacted by the experimental manipulation.

### Conclusion

Perceived stress does not impact proprioceptive accuracy.

## Introduction

Proprioception refers to the perception of the motor system (e.g., joint position, muscle contraction), based on the signals originating from mechanoreceptors located in muscles,

**Data availability statement:** The study has been preregistered at https://osf.io/ze5rs. The data and the analysis are available at: https://doi.org/10.6084/m9.figshare.28120619.

**Funding:** This work was supported by the Research Fund of the National Research, Development and Innovation Office (K 147788) and by the EKÖP-24 University Excellence Scholarship Program of The Ministry for

Culture and Innovation from the Source of The National Research, Development And Innovation Fund (Grant number: EKÖP-24-4-II-ELTE-543) and The funders had no role in study design, data collection and analysis, decision to publish, or preparation of the manuscript.

**Competing interests:** The authors have declared that no competing interests exist.

ligaments, joints, and skin surface, modified by central processes (e.g., the sense of effort) [1]. Proprioceptive accuracy (the ability to perceive proprioceptive information; PAc) shows important associations with motor performance [2,3]. The goal of the current study is to explore how acute stress affects PAc. As PAc is a fundamental ability in learning and executing motor skills, understanding how the stress response affects it can broaden our knowledge about the effect of stress on motor control and motor/sports performance.

The stress response prepares the organism to cope with external and internal physical and psychological challenges (i.e., stressors) [4]. The organism's general response consists of a wide range of physiological and psychological changes, for example, cortisol hormone is released [5]. and individuals show a significant elevation in heart rate (HR) [6]. In addition, heart rate variability (HRV) also changes; more specifically, indices associated with parasympathetic (vagal) impact on heart decrease [7]. Electrodermal activity (the constant change in skin electrical conductance caused by the activity of the eccrine sweat glands), can also be used as an arousal indicator because this response is controlled by the sympathetic nervous system [8]. Considering the psychological level, stressful situations increase perceived stress and state anxiety [9,10]. Interestingly, the aforementioned physiological and psychological aspects of acute stress often dissociate, i.e., do not correspond to each other [9,11,12].

In the laboratory, stress can be induced in different ways. Based on the review of Dickerson and Kemeny [11], psychological stress triggers the most intense rise in cortisol levels if participants face uncontrollability and social-evaluative threats. For this purpose, one of the most widely used tests is the Trier Social Stress Test [13]. However, this paradigm requires the presence of more experimenters, which limits the usability of the test [14]. To handle this issue, Almazrouei and colleagues [14] have developed and validated a stress paradigm (i.e., an online quiz involving general knowledge and mathematical questions, administered with and without time pressure and instant feedback on performance) that does not require the presence of an experimenter. According to their findings, the stressful version of the quiz caused a substantial increase in participants' state anxiety and perceived stress level compared to the control version. An additional goal of the current study was to develop and evaluate the Hungarian version of this test. In addition to self-reported stress, we also aimed to check if this paradigm elicits physiological stress (i.e., changes in cortisol, HR, HRV, and skin conductance level, SCL).

The exact mechanisms of how stress affects motor performance are only partially understood; multiple models have been proposed to explore this relationship. One of the first and best-known theories is the Yerkes-Dodson law, based on the observation of the behavior of mice in a discrimination task, stating that performance and arousal have an inverted U-shape relationship for difficult tasks and a linear positive relationship for simple tasks [15,16]. However, the model might have become overgeneralized [17]. For example, it may be valid for humans when learning new motor skills, but not when athletes execute skilled movements [18]. The exact type of motor skill in question also matters; high arousal has a negative effect mostly on movements requiring fine motor skills [19]. In addition, there appear to be individual differences in the response, i.e., some people might perform best when the stress level is high [20,21]. Recent reviews have concentrated on the possible role of attention [22], and conscious motor processing [23] in the anxiety-performance relationship, but no clear conclusion emerged. Nieuwenhuys and Oudejans [24,25] proposed an integrated model of how anxiety affects perceptual-motor performance, on the attentional, interpretational, and physical response levels. According to the model, anxiety draws attention to threat-related stimuli instead of task-relevant stimuli, leads to interpretation of ambiguous stimuli as threatening, and results in increased muscle tension and a tendency for avoidance behavior. Proprioceptive accuracy can deteriorate with allocating attention to another stimulus [26–28].

Also, an increased level of muscle tension might influence how proprioceptive stimulus is perceived [29], and the fact that extension is associated with avoidance behaviour, while flexion is associated with approach behavior [30] can cause additional noise. To date, there has been published only one study that investigated the relationship between stress and PAc [31]. The authors compared the performance of students in a relatively relaxed (i.e., one month before the committee exam) and stressful period (i.e., on the day of their final exam). The task assessed the accuracy of joint position repositioning in the ankle joint with eyes open (control condition) and eyes closed (proprioceptive condition). It was found that accuracy decreased in the stressful period in the proprioceptive, but not in the control condition, and the decrement in accuracy was positively associated with the increment in cortisol level. Findings of the study support the idea that the magnitude of the physiological stress response and PAc shows a negative, linear relationship. However, the study has serious methodological limitations: it applied a quasi-experimental design without a control group. Therefore, causal consequences could not have been drawn from the findings; for example, it is possible that differences in cortisol level and PAc can be explained by a third variable (e.g., increased cognitive load or bad sleep quality). Our primary goal was to replicate these findings with a more controlled, experimental design.

In conclusion, we hypothesized that laboratory-induced acute stress would have a negative impact on PAc. In addition, we intended to explore the associations between PAc and perceived and physiological stress.

## Methods

The study has been preregistered at https://osf.io/ze5rs.

### Participants

57 first-year university students participated in the study (86% female, $M_{age}$ = 20.4, $SD_{age}$ = 1.91). Sample size calculation was carried out using the G*power software [32] and was based on a 2 (group: experimental and control) * 2 (time: pre and post) mixed ANOVA, with α = 0.05, β = 0.95 and medium effect *(f = 0.25)* size based on the results of Şenol and colleagues [31]. The participation was voluntary and compensated with bonus point on a university course. The study was approved by the Research Ethics Committee of the Faculty of Education and Psychology at ELTE Eötvös Loránd University under the registration number of 2023/506. Participants signed a written informed consent form before the experiment, confirming that they were not treated for a neurological or psychiatric condition, and were not under the influence of alcohol or any psychoactive substances. Data was collected and recorded anonymously. The recruitment period for this study lasted from 17th of November 2023 to 7th of June 2024.

### Experimental manipulation

To induce stress, we adapted the procedure published by Almazrouei and colleagues [14] to the Hungarian language. Both groups were tasked with answering randomly selected mathematic and general knowledge questions. In the beginning, the stress group received a warning that their performance was being monitored. After that, if a participant answered incorrectly, a red "WRONG" message instantly appeared on the screen, while a grey "OK" message was displayed for correct answers, and if time ran out before a question was answered, a red "TIME OUT!" message was shown. The control group was asked to complete a similar set of questions in both number and type but without any feedback or time constraints. For more details, see the original publication [14].

## Measurements

**Proprioceptive accuracy (PAc).** The Joint Position Reproduction test (JPR) was used to assess PAc [33]. The position of the left elbow joint was moved and measured by a custom-made motorized device, with a precision of ±0.1 degree. The joint was moved passively from a starting position with a given speed to a target position and spent a given amount of time there. After that, the arm was moved back to the starting position and in the next step, participants had to reproduce the target position as accurately as possible. We used two different versions: in the active task participants had to actively move their arm in the repositioning phase, while in the passive condition, the joint was moved by the machine. In both versions, reaching the target position was signalled with a button press by the participant. There were 2 starting positions (30 and 150 degrees), 3 target positions (60, 90, 120 degrees), 2 movement speeds (12, 24 degrees/sec), and 2 amounts of time spent in the target position (2, 4 seconds). Each combination was presented once, resulting in 24 trials per condition. To assess participants' performance, we have used the absolute error score (i.e., the mean of the differences between target and reproduced positions), higher scores indicating worse PAc. Extreme outlying values (outside 3 interquartile range) were identified with boxplots per trial, considered artifacts, and removed from the analysis.

**Physiological stress.** Physiological measures (electrocardiography (ECG), and electrodermal activity) were recorded by the NeXus system (NeXus-10 Mark II, Version 1.02 device and BioTrace + Software for NeXus-10 v2018A1; Mind Media BV, Herten, the Netherlands). The ECG was recorded using the modified Lead II electrode arrangement (right clavicula and the lower part of the left ribcage). A notch filter was applied at 50Hz while the sampling rate was 1024/s. Electrodermal activity was assessed as skin conductance level (SCL), electrodes were placed on the palmar side of the second digit of the right index and middle fingers. The sampling frequency was set to 64/s. The skin was washed with soap and water before the attachment of the electrodes (d = 8mm); GEL101A isotonic electrode gel for EDA (BIOPAC Systems, Inc., Goleta, USA) was used. Possible artifacts were visually checked for all participants.

Tonic electrodermal activity (SCL) is considered a direct indicator of slow changes in sympathetic activation in response to emotional stimuli, including stressors [34,35]. After excluding the artifacts, the BioTrace + software was used to calculate mean SCL for the first 60 sec (pre) and final 60 sec period (post) of the quiz. As no ECG-based direct indicators of cardiac sympathetic activation exist, three indirect indicators, estimating parasympathetic impact on heart, were used. As a general rule, it was assumed that decreasing parasympathetic activity shows a shift from a more relaxed physiological state toward a more activated state. Under resting conditions, HR is controlled almost exclusively by the parasympathetic nervous system [36]; thus, an increase in HR in this domain indicates decreasing parasympathetic (vagal) activity. Similarly, as certain indices of HRV, such as the high-frequency domain of HRV (HF) and the root mean square of successive differences between normal heartbeats (RMSSD), correlate well with parasympathetic impact on the heart [36,37], decreasing values indicate decreasing parasympathetic activation. ECG data were analysed with the KubiosHRV Premium 3.5.0. software (Kubios Oy, Finland). The software automatically detects QRS-complexes, identifies and corrects for ectopic and missing heartbeats, and finally calculates the required indices [38,39]. RR interval detrending was carried out using the smoothn priors method with a smoothing parameter of 500. The level of automatic quality detection was set to medium, beat correction was automatic with an acceptance threshold of 5%.

Due to technical errors, ECG were missing for 10 participants and SCL was missing for 15 participants. In the preregistered study design, it was claimed that cortisol level will also be

assessed. However, due to a technical error, this measurement was not carried our properly thus it is not included in the study.

**Psychological stress.** The Hungarian state version [40] of the Spielberger State-Trait Anxiety Inventory (STAI-S) [41] was used to assess the current psychological stress level of the participants. The scale consists of 20 questions (e.g., "I am tense"), rated on a 4-point Likert scale from not at all (0) to fully (3) for each statement. A higher total score reflects higher perceived anxiety. Internal consistency values were excellent for both the pre (Cronbach's alpha = 0.92) and post measurement (Cronbach's alpha = 0.94)

## Procedure

When arriving at the laboratory, participants signed an informed consent form, and a research assistant applied electrodes and sensors to measure physiological activity (ECG, SCL). Afterwards, participants completed the baseline active and passive PAc tests in a randomized order, followed by the first completion of the STAI-S questionnaire. After initial measurements, participants fulfilled the online quiz; they were randomly assigned to the stressor or to the control group by the Qualtrics software. After the online quiz, participants filled out STAI-S again and completed the PAc tests.

## Statistical analysis

Statistical analysis was conducted with the jamowi v2.6.13.0 software [42]. SCL, HF, and RMSSD data were log-transformed to reduce deviation from normal distribution. Homogeneity of the groups with respect to age, and psychological and physiological variables at baseline was checked with independent samples t-tests and $\chi^2$-test (for gender ratio). For manipulation check, mixed analyses of variance (ANOVA) were used with psychological and physiological stress indices as outcome variables, time (pre- and post-intervention) as within-subject factor and group (control and stress group) as between-subject factor.

Hypotheses were checked with mixed frequentist and Bayesian ANOVA with JPR absolute error as outcome variable, time (pre- and post-intervention) as within-subject factor and group (control and stress group) as between-subject factor. For the Bayesian analyses, the null model included time, group, subjects, and random slope whereas the alternative model represented the time*group interaction. For the explorative part of the study, frequentist and Bayesian Pearson correlations were used. For the Bayesian analyses, Bayes factors ($BF_{10}$) above 3 were considered supporting the alternative model, and Bayes factors below .33 were regarded as supporting the null model. The range between .33 and 3 were considered inconclusive [43].

## Results

### Descriptive statistics, baseline comparisons and manipulation checks

Descriptive statistics of the assessed variables are presented in Table 1. The two groups showed no significant differences with respect to age ($t(55) = -0.251$; $p = .803$; $d = -0.066$) and gender ratio ($\chi^2(1) = 0.029$, $p = .957$).

For STAI-S, independent-samples t-test indicated no significant differences between the control and stress group at baseline ($t(55) = -0.506$; $p = .615$; $d = -0.134$). Repeated-measures ANOVA showed a significant time main effect ($F(1,55) = 18.048$; $p < .001$; $\eta^2 = 0.04$) with a small effect size, a significant group main effect ($F(1,55) = 9.58$; $p = .003$; $\eta^2 = 0.113$) with a medium effect size, and a significant time*group interaction ($F(1,55) = 34.1$; $p < .001$; $\eta^2 = 0.076$) with a medium effect size. The STAI-S score significantly increased in the experimental but not in the control group after the task (Fig 1).

**Table 1. Descriptive statistics (M ± SD) of the assessed variables split by group and time of assessment.**

| Variable | Control group (n = 28) | | Stress group (n = 29) | |
|---|---|---|---|---|
| | Pre | Post | Pre | Post |
| **Proprioceptive accuracy** | | | | |
| PAc active (°) | 5.66 ± 0.97 | 5.98 ± 1.83 | 5.92 ± 1.18 | 5.77 ± 1.58 |
| PAc passive (°) | 6.0 ± 1.34 | 5.88 ± 1.59 | 6.31 ± 1.42 | 5.98 ± 1.41 |
| **Psychological stress** | | | | |
| STAI-S | 17.5 ± 9.3 | 16.0 ± 8.3 | 18.7 ± 9.0 | 28.5 ± 9.9 |
| **Physiological stress** | | | | |
| HR (*bpm*) | 78.3 ± 12.89 | 82.9 ± 12.1 | 77.82 ± 7.57 | 84.06 ± 8.7 |
| HF (*log ms²*) | 2.95 ± 0.4 | 2.78 ± 0.48 | 2.86 ± 0.32 | 2.68 ± 0.37 |
| RMSSD (*log ms*) | 1.66 ± 0.22 | 1.59 ± 0.23 | 1.63 ± 0.14 | 1.53 ± 0.15 |
| SCL (*log µS*) | 0.67 ± 0.3 | 0.6 ± 0.26 | 0.66 ± 0.3 | 0.61 ± 0.27 |

STAI-S: Spielberger State-Trait Anxiety questionnaire state version; PAc: proprioceptive accuracy; HR: heart rate; HF: High frequency domain of heart rate variability; RMSSD: root mean square of successive differences between normal heartbeats; SCL: skin conductance level.

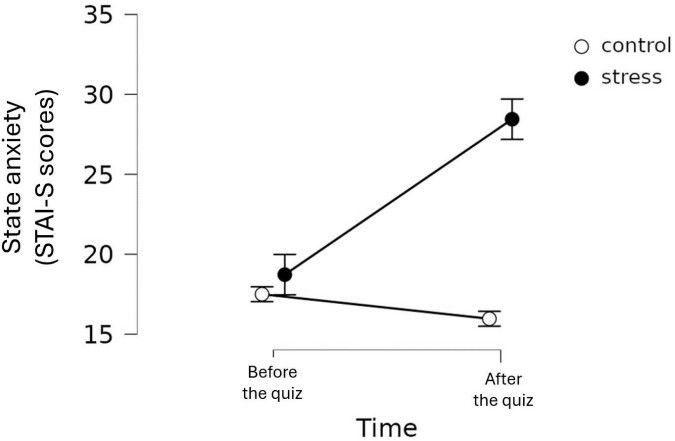

**Fig 1. Changes of state anxiety (measured by STAI-S) in the two groups before and after completing the quiz (error bars represent standard errors).**

For HR, independent-samples t-test indicated no significant differences between the control and stress group at baseline ($t(46) = 0.164$; $p = .871$; $d = 0.048$). Repeated-measures ANOVA showed a significant time main effect ($F(1,46) = 45.71$; $p < .001$; $\eta^2 = 0.067$) with a medium effect size, indicating that HR significantly increased in both groups. No significant group main effect ($F(1,46) = 0.014$; $p = .907$; $\eta^2 < .001$), and no significant time*group interaction ($F(1,46) = 1.05$; $p = .311$; $\eta^2 = 0.002$) was found.

For HF, independent-samples t-test indicated no significant differences between the control and stress group at baseline ($t(46) = 0.875$; $p = .386$; $d = 0.256$). Repeated-measures ANOVA showed a significant time main effect ($F(1,46) = 13.674$; $p < .001$; $\eta^2 = 0.049$) with small effect size, showing that HF significantly decreased in both groups after the intervention. No significant group main effect ($F(1,46) = 0.918$; $p = .343$; $\eta^2 = 0.015$), and no significant time*group interaction ($F(1,46) = 0.029$; $p = .866$; $\eta^2 < .001$) was revealed.

For RMSSD, independent-samples t-test indicated no significant differences between the control and stress group at baseline ($t(46) = 0.519$; $p = .606$; $d = 0.152$). Repeated-measures ANOVA showed a significant time main effect ($F(1,46) = 23.601$; $p < .001$; $\eta^2 = 0.054$) with small effect size, meaning that RMSSD significantly decreased in both groups. No significant group main effect ($F(1,46) = 0.681$; $p = .414$; $\eta^2 = 0.012$), and no significant time*group interaction ($F(1,46) = 0.635$; $p = .43$; $\eta^2 = .001$) were found.

For SCL, independent-samples t-test indicated no significant differences between the control and stress group at baseline ($t(40) = 0.871$; $p = .389$; $d = 0.270$). Repeated-measures ANOVA showed no significant time main effect ($F(1,40) = 0.008$; $p = .930$; $\eta^2 < .001$), no significant group main effect ($F(1,40) = 0.528$; $p = .472$; $\eta^2 = 0.013$), and no significant time*group interaction ($F(1,40) = 3.267$; $p = .078$; $\eta^2 < .001$).

## Hypothesis tests

For the active version of PAc, independent-samples t-test indicated no significant differences between the control and stress group at baseline ($t(55) = -0.889$; $p = .378$; $d = -0.236$). Repeated-measures ANOVA showed no significant time main effect ($F(1,55) = 0.186$; $p = .668$; $\eta^2 < .001$), no significant group main effect ($F(1,55) = 0.005$; $p = .942$; $\eta^2 < .001$), and no significant time*group interaction ($F(1,55) = 1.411$; $p = .240$; $\eta^2 = 0.007$) (Fig 2). Bayesian repeated-measures ANOVA indicated a Bayes-factor in the inconclusive domain for the time*group interaction ($BF_{10} = 0.485$).

For the passive version of PAc, independent-samples t-test indicated no significant differences between the control and stress group at baseline ($t(55) = -0.848$; $p = .400$; $d = -0.225$). Repeated-measures ANOVA showed no significant time main effect ($F(1,55) = 1.237$; $p = .271$; $\eta^2 = .01$), no significant group main effect ($F(1,55) = 0.398$; $p = .531$; $\eta^2 = .005$), and no significant time*group interaction ($F(1,55) = 0.276$; $p = .601$; $\eta^2 = 0.001$) (Fig 2). Bayesian repeated-measures ANOVA indicated the superiority of the null hypothesis for the time*group interaction (i.e., lack of difference; $BF_{10} = 0.302$).

## Explorative analysis

Although no group-level differences in PAc were found, we carried out a within-subject analysis, exploring the associations between PAc and physiological and psychological indicators aspects of the stress response. As for the associations between the two versions of PAc and psychological and physiological indicators of the stress response, frequentist analysis did not indicate significant correlations (for details, see Table 2). Bayesian analysis suggested the superiority of the null hypothesis for the majority of the analyses (8 of 10) and was indecisive for the remaining correlations.

## Discussion

In an experimental study with the participation of 57 young individuals, laboratory-induced psychological stress did not impact PAc as assessed with the active and passive versions of the joint reproduction test for the elbow joint. Furthermore, changes in PAc were not related to changes in state anxiety and physiological indicators of the acute stress response, such as changes in heart rate, heart rate variability, and skin conductance level.

To induce a stress response, we have adapted the stress paradigm developed by Almazrouei and colleagues [14]. Similarly to the original study, we found a significant increase in state anxiety (i.e., perceived stress) for the stressed group, compared to the control group. However, the experimental manipulation did not affect the assumed stress-related physiological changes. HR showed a significant increase, whereas cardiac indicators of parasympathetic

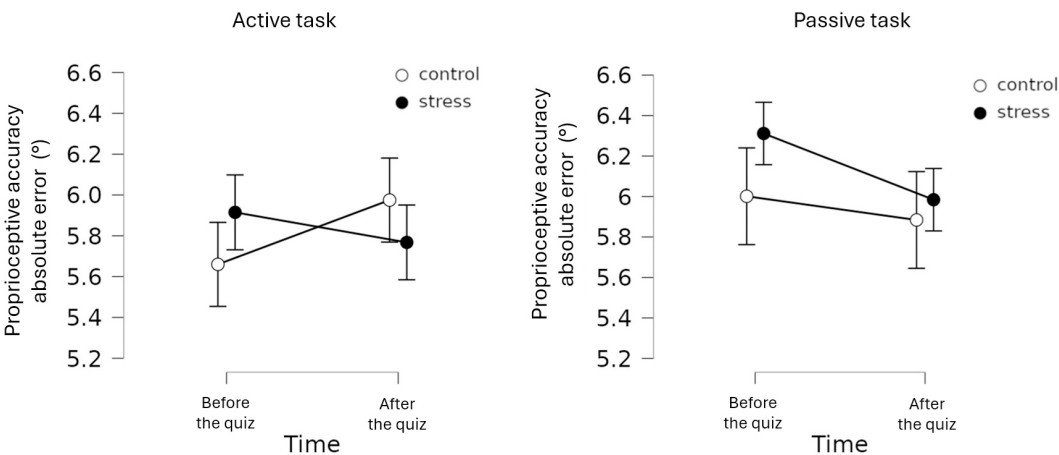

**Fig 2. Changes of Proprioceptive Accuracy indices measured by the Joint Position Reproduction test (JPR) in the two groups (error bars represent standard errors).**

**Table 2. Results of frequentist and Bayesian correlation analysis exploring the associations between active and passive versions of PAc and various indicators of the stress response (Pearson correlation coefficients).**

| | PAc active change | *p* | *BF10* | PAc passive change | *p* | *BF10* |
|---|---|---|---|---|---|---|
| **STAI-S change** | −.117 | .388 | 0.238 | −.073 | .587 | 0.191 |
| **HR change** | −.119 | .420 | 0.247 | −.212 | .148 | 0.497 |
| **HF change** | .102 | .940 | 0.227 | .032 | .830 | 0.184 |
| **RMSSD change** | .088 | .552 | 0.214 | −.043 | .769 | 0.188 |
| **SCL change** | −.253 | .106 | 0.682 | .002 | .989 | 0.192 |

STAI-S: Spielberger State-Trait Anxiety questionnaire state version; PAc: proprioceptive accuracy; HR: heart rate; HF: High frequency domain of heart rate variability; RMSSD: root mean square of successive differences between normal heartbeats; SCL: skin conductance level.

activation (HR, RMSSD) uniformly decreased in both groups during the task. The direction of these changes is in accordance with the idea that parasympathetic deactivation/sympathetic activation took place during both the stressful and control versions of the quiz. As a substantial difference in perceived stress level was found, the lack of difference with respect to the physiological indicators of the stress response can be explained by the well-known dissociation between bodily changes and their perceived counterparts (for a review, see [44]). In addition, individual differences in the physiological stress response, e.g., cardiovascular reactivity, might have blurred the group-level differences.

Despite the substantial increase of perceived stress, the experimental manipulation did not decrease PAc, as assessed with the active and passive versions of the Joint Position Reproduction test for the elbow joint. Thus, we could not replicate the results of Şenol and colleagues [31]. There are multiple possible explanations for this contradiction. An important difference between the two studies is that the current one used an experimental setup and a true randomized design, while the original utilized a quasi-experimental design. The lack of control group and randomization in the study of Şenol and colleagues [31] means that a third variable might have caused the reported significant association. So, the null finding can reflect the lack of effect in the population, meaning that proprioceptive accuracy does not deteriorate with acute psychological stress. From a practical point of view, this would mean that when measuring proprioceptive accuracy, for example in sports or rehabilitation context,

one does not have to worry about an acute psychological stressor that would influence the results before the measurement. Also, as proprioceptive accuracy might mostly play a role in the initial (cognitive) phase of motor skill acquisition, an acute psychological stressor would not disrupt this process. A possible explanation for the null findings in the context of the integrated model of perceptual-motor performance [24], is that the effect of anxiety might be compensated with different strategies. For example, with increased mental effort, people can inhibit anxiety-==elated stimuli and concentrate on task-relevant stimuli. Also, people might use anxiety-reducing strategies. It is possible that participants of this study could successfully use these strategies. However, there are alternative explanations. For example, another difference between the Şenol study and our study is that we assessed the immediate effect of an acute stressor, while the assumed stressor (i.e., preparation for a final exam) might have been more prolonged in the original study [31]. It is possible that proprioceptive accuracy is only affected if the stress response is more prolonged.

A further option is that the decrement in PAc is related to the physiological aspect of the stress response, not to the perceived aspect. For example, it is possible that increased internal noise due sympathetic activation has a negative impact on the low-level processing of the proprioceptive signal. Results of the explorative part of the study, revealing no associations between participants' performance and any physiological change, however, do not support this idea. But it also cannot be excluded: as the manipulation was not successful in inducing a physiological stress response, the variability in the sample might be too low to show this effect.

Another important point to discuss is that when evaluating the results related to proprioception, it is always important to consider the joint measured, and the method used [2]. For this study, we used the Joint Position Reproduction test with active and passive versions for the right elbow joint, while Şenol and colleagues [31] used the Joint Position Reproduction test for the ankle joint only with active reproduction. It cannot be excluded that stress influences the accuracy shown in different body sites differently.

## Limitations and future directions

An important limitation of the current study is that the sample was homogeneous as the study involved first year university students, 86% of whom were female. This limits the generalizability of the findings. For future studies, it is recommended to use a more diverse, preferably representative sample. It is possible that the stress effect on proprioceptive accuracy is not linear and shows high variability across individuals. Future studies should also test these possibilities.

Also, the ecological validity of the quiz-based stressor used in this study is limited as it represents a test or exam like acute stress which may have smaller or different effects than a real-life stressor, not to speak of longer-term chronic stress. Moreover, it is not clear how long the psychological stress induced by our stress paradigm lasts. Completing the active and passive versions of the PAc test in the study took approximately 20 minutes. Thus, it is possible that participants' perceived stress levels substantially decreased by the end of the second PAc test. Dividing attention can decrease PAc [26]; thus, if one administers a stress task simultaneously, it won't be clear that an eventual decrease in PAc should be interpreted as the result of the stress or the result of the decreased cognitive capacity. Therefore, a necessary step for future studies would be the development of stress paradigms that can be administered before the PAc test(s) and that have sufficiently long-lasting psychological effects. In line with that, future studies should test the effect of chronic stress on PAc. Additionally, not just cognitive stress, but also emotional aspects or physical stress induction could be used in the future, in order to explore the differences in the impact of different aspects of stress.

The study focused only the elbow joint, so future studies could expand the proprioception related measures. In the future, either weight discrimination tests or tests of balancing ability could be included, and consideration should be given to measuring the accuracy of other joints.

Technical issues that lead to the loss of physiological data (ECG and SCL) may reduce the statistical power and potentially impact the accuracy and reliability of the findings of the present study. These problems may have contributed to inconclusive findings. Also, the sample size calculation was based on a medium effect size, and it is possible that the effect size is smaller, and more participants would be needed to show the hypothesized effects. Overall, avoiding data loss, and increasing sample size can increase the reliability of the findings.

## Conclusion

The stress-inducing method of Almazrouei and colleagues [14] caused a significant increase in psychological stress (anxiety) but did not evoke physiological stress. This psychological stress had no effect on participants' proprioceptive accuracy. Changes in proprioceptive accuracy were not associated with psychological and physiological aspects of the stress response. Future research could focus not only acute but on chronic stress, or could use multiple stress-induction methods, and more proprioceptive tasks to better understand the relationship between stress and proprioceptive accuracy.

## Author contributions

**Conceptualization:** Áron Horváth.

**Data curation:** Blanka Aranyossy, Áron Horváth.

**Formal analysis:** Ferenc Köteles.

**Funding acquisition:** Ferenc Köteles.

**Investigation:** Blanka Aranyossy.

**Methodology:** Adam Koncz, Ferenc Köteles, Áron Horváth.

**Project administration:** Blanka Aranyossy.

**Supervision:** Ferenc Köteles.

**Writing – original draft:** Adam Koncz, Ferenc Köteles, Blanka Aranyossy, Áron Horváth.

**Writing – review & editing:** Adam Koncz, Ferenc Köteles, Áron Horváth.

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
