## [Decision Letter · Decision Letter 0]

17 Dec 2024

PONE-D-24-44997Acute psychological stress does not influence joint position reproduction performance in the elbow jointPLOS ONE

Dear Dr. Koncz,

Thank you for submitting your manuscript to PLOS ONE. After careful consideration, we feel that it has merit but does not fully meet PLOS ONE’s publication criteria as it currently stands. Therefore, we invite you to submit a revised version of the manuscript that addresses the points raised during the review process.

**ACADEMIC EDITOR: **Dear Authors

The manuscript underwent evaluation by two expert reviewers and an academic editor. Both reviewers concluded that the manuscript is well-written and addresses a significant research area. I believe that the manuscript will be significantly improved after revision.

Comments:

1. Authors should address the limitations of the sample, which consists only of young university students.

2. Numerous repetitions are evident, especially in the discussion section.

3. Remove citations from the abstract.

4. Authors should explain their rationale for selecting only heart rate (HR), high frequency (HF) of heart rate variability (HRV), and root mean square of successive differences (RMSSD) from ECG data.

5. In electrodermal activity, skin conductance response (SCR) and skin conductance level (SCL) are frequently utilized. Nonetheless, the present study exclusively employed SCL. Kindly provide a rationale.

6. Authors must present a comprehensive methodology for the preprocessing of ECG and EDA data.

Please revise your manuscript based on my comments and other comments given by both the reviewers.

We look forward to receiving your revised manuscript.

Kind regards,

Shahnawaz Anwer, PhD

Academic Editor

PLOS ONE

Journal Requirements:

2. Thank you for stating the following financial disclosure: [FK received funding from Research Fund of the National Research, Development and Innovation Office (K 147788),

AK received funding by the EKÖP-24 University Excellence Scholarship Program of The Ministry for Culture and Innovation from the Source of The National Research, Development And Innovation Fund (Grant number: EKÖP-24-4-II-ELTE-543)].

3. Thank you for uploading your study's underlying data set. Unfortunately, the repository you have noted in your Data Availability statement does not qualify as an acceptable data repository according to PLOS's standards.

Reviewers' comments:

Reviewer's Responses to Questions

**Comments to the Author**

1. Is the manuscript technically sound, and do the data support the conclusions?

Reviewer #1: Partly

Reviewer #2: Partly

2. Has the statistical analysis been performed appropriately and rigorously? 

Reviewer #1: Yes

Reviewer #2: Yes

3. Have the authors made all data underlying the findings in their manuscript fully available?

Reviewer #1: Yes

Reviewer #2: Yes

4. Is the manuscript presented in an intelligible fashion and written in standard English?

Reviewer #1: Yes

Reviewer #2: Yes

5. Review Comments to the Author

Reviewer #1: Article theme and research question:

The article clearly defines its theme, focusing on the impact of acute stress on individual proprioceptive accuracy (PAc). This research question holds significant importance in the fields of exercise and neuroscience.

Literature review:

The literature review section generally covers research results in related fields, but it lacks references to some of the latest or key studies, which may limit readers' comprehensive understanding of the research background. It is suggested that the author should include more recent literature directly related to this study in the revision, to enhance the timeliness and depth of the review.

Sample Selection

The sample primarily comprises first-year college students, with a disproportionately high number of females, potentially compromising the representativeness of the sample. It is advised that the author considers a broader sample base in subsequent research to improve the study's generalizability and applicability.

Data Collection and Analysis: The methods of data collection are generally reasonable, albeit with some technical challenges, such as the loss of certain physiological data (ECG and SCL), which could potentially impact the accuracy and reliability of the findings. Of course, the author can incorporate age groups and gender as subgroups in the study later on to see if different results would be obtained.

Research Results:

The research outcomes are largely lucid, albeit with some ambiguous expressions or inadequate explanations. For instance, the elucidation of certain statistical results (e.g., non-significant outcomes) is inadequate, lacking exploration into potential causes or implications. The intriguing aspect of how to induce psychological stress and simultaneously manage physiological responses between the experimental and control groups deserves a detailed elucidation from the author. Additionally, it is pertinent to inquire whether the timing of acute stress exposure impacts the results, specifically whether there are differences in proprioceptive accuracy at 10 seconds, 30 seconds, 1 minute, and 5 minutes following acute psychological stress. It is recommended that the author conduct a detailed analysis of this matter. Moreover, there exists a discernible disparity between the research findings and the anticipated objectives, specifically, acute stress did not significantly impact proprioceptive accuracy. The author ought to undertake a more comprehensive analysis and discussion to pinpoint research limitations and future research trajectories.

Discussion and Conclusion

The discussion section primarily covers the significance and limitations of the research findings, yet it lacks an in-depth exploration of certain key issues. For instance, the author fails to adequately explain why acute stress did not affect proprioceptive accuracy, and how this finding relates to existing literature. The conclusion section is relatively concise, yet it lacks clear recommendations or directions for future research. It is suggested that the author add more prospects and suggestions for future research in the conclusion section to guide further development in this field. In summary, this article possesses certain theoretical and practical significance in exploring the impact of acute stress on proprioceptive accuracy. The overall language of the article is fluent, and the chart production is generally standardized. However, there are still some shortcomings in research methods, sample selection, data collection and analysis, interpretation of research results, and discussion and conclusions. It is recommended that the author can consider the aforementioned suggestions during revision.

Reviewer #2: Thanks for receiving the opportunity to review your paper. Overall, I appreciate the thoroughness and importance of your study on acute psychological stress does not influence joint position reproduction performance in the elbow joint. However, this manuscript must undergo minor revisions before it can be considered for publication.

1. The abstract is well-structured but could benefit from a more concise summary of key results. the keyword should in MeSH terms. The results section should reflect the statistical values of the outcomes

2. The introduction is comprehensive but could be streamlined to avoid redundancy, particularly in the explanation of stress mechanisms and proprioception.

3. The double-blind design and randomization are strong aspects of the methodology. However, the limitations of using a homogeneous sample (primarily young university students) should be emphasized more prominently in the discussion.

4. The reliance on self-reported stress data and the loss of physiological data due to technical issues (e.g., ECG and SCL) weaken the reliability of the findings. Future studies should address these issues to strengthen conclusions.

5. The results indicate inconclusive findings for some measures. Adding a discussion on why these results were inconclusive (e.g., sample size, experimental design) would enhance the reader's understanding.

6. The discussion effectively highlights the discrepancy between perceived stress and physiological responses. However, further exploration of how these results fit within existing literature (beyond Şenol et al.) would provide more depth. The lack of effect on proprioceptive accuracy is explained well, but implications for broader contexts (e.g., sports, rehabilitation) could be expanded.

7. The tables are clear, but the presentation of descriptive statistics (e.g., in Table 1) could be visually improved with better formatting.

8. Figures are informative, though labels and legends could be enhanced for clarity

9. While limitations are acknowledged, they could be more critically analyzed, particularly regarding ecological validity and the generalizability of findings.

10. Future research directions should include testing a more diverse population and incorporating chronic stress models.

6. PLOS authors have the option to publish the peer review history of their article (what does this mean?). If published, this will include your full peer review and any attached files.

Reviewer #1: No

Reviewer #2: No

---

## [Author Response · Author response to Decision Letter 1]

14 Jan 2025

Dear Dr. Koncz,

Thank you for submitting your manuscript to PLOS ONE. After careful consideration, we feel that it has merit but does not fully meet PLOS ONE’s publication criteria as it currently stands. Therefore, we invite you to submit a revised version of the manuscript that addresses the points raised during the review process.

ACADEMIC EDITOR:

Dear Authors

The manuscript underwent evaluation by two expert reviewers and an academic editor. Both reviewers concluded that the manuscript is well-written and addresses a significant research area. I believe that the manuscript will be significantly improved after revision.

Dear Editor

Thank you for the positive feedback. We addressed all suggestions and remarks; our responses to the comments can be found below.

Comments:

1. Authors should address the limitations of the sample, which consists only of young university students.

Thank you for this comment, the limitation section is extended by the following sentence: An important limitation of the current study is that the sample was homogeneous as the study involved first year university students, 86% of whom were female. This limits the generalizability of the findings. For future studies, it is recommended to use a more diverse, preferably representative sample. It is possible that the stress effect on proprioceptive accuracy is not linear and shows high variability across individuals. Future studies should also test these possibilities.

2. Numerous repetitions are evident, especially in the discussion section.

Thank you. The authors have tried to eliminate repetitions in the manuscript

3. Remove citations from the abstract.

We have removed the references from the abstract as suggested

4. Authors should explain their rationale for selecting only heart rate (HR), high frequency (HF) of heart rate variability (HRV), and root mean square of successive differences (RMSSD) from ECG data.

Thank you for this suggestion, the rationale for selecting these variables can now be found in the measurements section, please see the following sentences: “Under resting conditions, HR is controlled almost exclusively by the parasympathetic nervous system [36]; thus, an increase in HR in this domain indicates decreasing parasympathetic (vagal) activity. Similarly, as certain indices of HRV, such as the high-frequency domain of HRV (HF) and the root mean square of successive differences between normal heartbeats (RMSSD), correlate well with parasympathetic impact on the heart [36,37], decreasing values indicate decreasing parasympathetic activation.”

5. In electrodermal activity, skin conductance response (SCR) and skin conductance level (SCL) are frequently utilized. Nonetheless, the present study exclusively employed SCL. Kindly provide a rationale.

As we were curious about tonic changes of the sympathetic activation and not about acute responses to individual stimuli, SCL was the obvious choice. This is explained in the revised version.

6. Authors must present a comprehensive methodology for the preprocessing of ECG and EDA data.

Data processing is explained in the measurement section. For SCL: “Possible artifacts were visually checked for all participants. “ and “After excluding the artifacts, the BioTrace+ software was used to calculate mean SCL for the first 60 sec (pre) and final 60 sec period (post) of the quiz”.

In case of ECG please see the following sentence: “ ECG data were analysed with the KubiosHRV Premium 3.5.0. software (Kubios Oy, Finland). The software automatically detects QRS-complexes, identifies and corrects for ectopic and missing heartbeats, and finally calculates the required indices (Lipponen & Tarvainen, 2019; Tarvainen et al., 2014). RR interval detrending was carried out using the smoothn priors method with a smoothing parameter of 500. The level of automatic quality detection was set to medium, beat correction was automatic with an acceptance threshold of 5%.”

In the statistical analysis section, we mention that “SCL, HF, and RMSSD data were log-transformed to reduce deviation from normal distribution”

Please revise your manuscript based on my comments and other comments given by both the reviewers.

We look forward to receiving your revised manuscript.

Kind regards,

Shahnawaz Anwer, PhD

Academic Editor

PLOS ONE

Journal Requirements:

2. Thank you for stating the following financial disclosure: [FK received funding from Research Fund of the National Research, Development and Innovation Office (K 147788),

AK received funding by the EKÖP-24 University Excellence Scholarship Program of The Ministry for Culture and Innovation from the Source of The National Research, Development And Innovation Fund (Grant number: EKÖP-24-4-II-ELTE-543)].

We modified the cover letter and put in the following: The funders had no role in study design, data collection and analysis, decision to publish, or preparation of the manuscript.

3. Thank you for uploading your study's underlying data set. Unfortunately, the repository you have noted in your Data Availability statement does not qualify as an acceptable data repository according to PLOS's standards.

We have uploaded the data and analysis to Figshare, and also modified the doi in the manuscript (10.6084/m9.figshare.28120619)

Reviewers' comments:

Reviewer's Responses to Questions

Comments to the Author

1. Is the manuscript technically sound, and do the data support the conclusions?

Reviewer #1: Partly

Reviewer #2: Partly

2. Has the statistical analysis been performed appropriately and rigorously?

Reviewer #1: Yes

Reviewer #2: Yes

3. Have the authors made all data underlying the findings in their manuscript fully available?

Reviewer #1: Yes

Reviewer #2: Yes

4. Is the manuscript presented in an intelligible fashion and written in standard English?

Reviewer #1: Yes

Reviewer #2: Yes

5. Review Comments to the Author

Reviewer #1: Article theme and research question:

The article clearly defines its theme, focusing on the impact of acute stress on individual proprioceptive accuracy (PAc). This research question holds significant importance in the fields of exercise and neuroscience.

Literature review:

The literature review section generally covers research results in related fields, but it lacks references to some of the latest or key studies, which may limit readers' comprehensive understanding of the research background. It is suggested that the author should include more recent literature directly related to this study in the revision, to enhance the timeliness and depth of the review.

Thank you for this suggestion, we added more recent literature in the light of the integrated model of the effect of anxiety on perceptual-motor performance to the introduction section. please see: “Nieuwenhuys and Oudejans [24,25] proposed an integrated model of how anxiety affects perceptual-motor performance, on the attentional, interpretational, and physical response levels. According to the model, anxiety draws attention to threat-related stimuli instead of task-relevant stimuli, leads to interpretation of ambiguous stimuli as threatening, and results in increased muscle tension and a tendency for avoidance behavior. Proprioceptive accuracy can deteriorate with allocating attention to another stimulus [26–28]. Also, an increased level of muscle tension might influence how proprioceptive stimulus is perceived [29], and the fact that extension is associated with avoidance.”

Sample Selection

The sample primarily comprises first-year college students, with a disproportionately high number of females, potentially compromising the representativeness of the sample. It is advised that the author considers a broader sample base in subsequent research to improve the study's generalizability and applicability.

Thank you for this suggestion, the authors will try to pay more attention to the representativeness of the sample in the future. We explained the considerations with the sample more broadly in the limitations and further directions section.

Data Collection and Analysis: The methods of data collection are generally reasonable, albeit with some technical challenges, such as the loss of certain physiological data (ECG and SCL), which could potentially impact the accuracy and reliability of the findings. Of course, the author can incorporate age groups and gender as subgroups in the study later on to see if different results would be obtained.

Thank you, the authors totally agree and mentions that “Technical issues that lead to the loss of physiological data (ECG and SCL) may reduce the statistical power and potentially impact the accuracy and reliability of the findings of the present study. These problems may have contributed to inconclusive findings” in the limitations section

Research Results:

The research outcomes are largely lucid, albeit with some ambiguous expressions or inadequate explanations. For instance, the elucidation of certain statistical results (e.g., non-significant outcomes) is inadequate, lacking exploration into potential causes or implications. The intriguing aspect of how to induce psychological stress and simultaneously manage physiological responses between the experimental and control groups deserves a detailed elucidation from the author.

Thank you for the feedback; we have revised the manuscript to address these concerns by providing more detailed explanations. Please see the following: “Despite the substantial increase of perceived stress, the experimental manipulation did not decrease PAc, as assessed with the active and passive versions of the Joint Position Reproduction test for the elbow joint. Thus, we could not replicate the results of Şenol and colleagues [31]. There are multiple possible explanations for this contradiction. An important difference between the two studies is that the current one used an experimental setup and a true randomized design, while the original utilized a quasi-experimental design. The lack of control group and randomization in the study of Şenol and colleagues [31] means that a third variable might have caused the reported significant association. So, the null finding can reflect the lack of effect in the population, meaning that proprioceptive accuracy does not deteriorate with acute psychological stress. From a practical point of view, this would mean that when measuring proprioceptive accuracy, for example in sports or rehabilitation context, one does not have to worry about an acute psychological stressor that would influence the results before the measurement. Also, as proprioceptive accuracy might mostly play a role in the initial (cognitive) phase of motor skill acquisition, an acute psychological stressor would not disrupt this process. A possible explanation for the null findings in the context of the integrated model of perceptual-motor performance [24], is that the effect of anxiety might be compensated with different strategies. For example, with increased mental effort, people can inhibit anxiety-related stimuli and concentrate on task-relevant stimuli. Also, people might use anxiety-reducing strategies. It is possible that participants of this study could successfully use these strategies.”

Additionally, it is pertinent to inquire whether the timing of acute stress exposure impacts the results, specifically whether there are differences in proprioceptive accuracy at 10 seconds, 30 seconds, 1 minute, and 5 minutes following acute psychological stress. It is recommended that the author conduct a detailed analysis of this matter.

Reply: Unfortunately, the PAc test needs to be administer

---

## [Editor Report · Decision Letter 1]

28 Jan 2025

Acute psychological stress does not influence joint position reproduction performance in the elbow joint

PONE-D-24-44997R1

Dear Dr. Koncz,

We’re pleased to inform you that your manuscript has been judged scientifically suitable for publication and will be formally accepted for publication once it meets all outstanding technical requirements.

Kind regards,

Shahnawaz Anwer, PhD

Academic Editor

PLOS ONE
---

## [Editor Report · Acceptance letter]

PONE-D-24-44997R1

PLOS ONE

Dear Dr. Koncz,

I'm pleased to inform you that your manuscript has been deemed suitable for publication in PLOS ONE. Congratulations! Your manuscript is now being handed over to our production team.

Kind regards,

on behalf of

Dr. Shahnawaz Anwer

Academic Editor

PLOS ONE